# The Importance of the Geometry of the Down Sprue in the Gravity Casting Process

**DOI:** 10.3390/ma15144937

**Published:** 2022-07-15

**Authors:** Rafał Dojka, Jan Jezierski, Michał Szucki

**Affiliations:** 1ODLEWNIA RAFAMET Sp. z o.o., 1 Staszica, Kuźnia, 47-420 Raciborska, Poland; r.dojka@odlewnia-rafamet.pl; 2Department of Foundry Engineering, Silesian University of Technology, 7 Towarowa, 44-100 Gliwice, Poland; 3Foundry Institute, TU Bergakademie Freiberg, Bernhard-von-Cottastr. 4, 09599 Freiberg, Germany; michal.szucki@gi.tu-freiberg.de

**Keywords:** gating system, pouring process, down sprue, gas entrapment, sand mold

## Abstract

This article presents the results of experiments on the optimization of down sprue geometry in the process of pouring sand molds. Theoretical assumptions and computer simulation tests are presented. The starting point was the theory and experience of gas entrapment caused mainly by a poorly designed gating system and the down sprue. Simulations were performed using Magmasoft software. First, initial studies were carried out to determine how the geometry (mainly the channel cross-section) of the sprue affects the problem, and then a detailed experiment was carried out on the so-called ‘short sprue’ version. The air entrapment process was analyzed, as were the parameters of the liquid alloy flow that passes through the analyzed channels. Nine geometric versions of the sprue were proposed and analyzed, and the results allowed us to conclude which sprue geometry is the best from the point of view of minimization of the gas entrapment problem.

## 1. Introduction

There is no doubt that the quality of castings is highly dependent on the gating system. This is true for every foundry technology and every type of casting mold used. One of the key factors that affect the internal quality of the casting is the gas (mainly air) during the pouring of the mold [1,2,3,4]. It is very often caused by a poorly designed gating system, mainly the pouring basin (a cup) and the down sprue (sprue), which together create the inlet for the liquid alloy going into and then through the casting mold. Too much air trapped (sucked in) by the initial parts of the gating system often results in major gas porosity defects and the appearance of so-called bifilms that typically scrape the casting [1,2,3,5,6]. Although this is quite common knowledge, there is still a huge discussion of whether it is technologically and economically justified to design a gating system in order to minimize the air entrapment process. The authors, knowing the theories of Professor John Campbell and their further development by others [7,8,9] well and having their own experiences on this issue [10,11,12], carried out a complex experimental plan to thoroughly discuss the importance of sprue geometry.

Often, in foundry engineering practice, during the pouring of the mold, the situation presented in Figure 1a arises. Poorly designed geometry and, in consequence, the excess volume of the sprue favor gas entrainment, and this is an excellent way to deteriorate the quality of the liquid alloy. Such an unacceptable shape of the sprue can be a result of either an improper tapering angle or the utilization of commercially available refractory tubing for sprue construction. In such sprues, after the passing of the initial portion of the metal, the sprue will gradually fill from the bottom to the top, as shown in Figure 1a. At the currently filled level of the sprue, entrainment of bifilms and bubbles will take place, which was presented in [13] and is shown in Figure 1b. Additionally, in the sprue, underpressure occurs, which surrounds the falling stream, creating a perfect environment for sucking gases from the mold into the flowing melt [1,14,15]. It should be mentioned that this back wave, which travels up the sprue after reaching the pouring basin filled with metal, may create splashes that are dangerous for the foundry crew.

Unfortunately, the phenomenon presented in Figure 1 is a common practice, as many foundries and even scientists manufacture their castings in this way [17,18,19,20], which are presented in Figure 2. The sprues shown in Figure 2b,c cause entrainment and also dramatically decrease the yield. In some cases, the excessive volume of the sprue in the bottom gating may cause more entrainment than in the case of the top gating, which was proven in [13]. These results should lead to serious thought by foundrymen that, regardless of the design, the bottom gating is sufficient to obtain a sound casting.

In [20], the authors simulated the initial mold-filling phase for the casting of steel ingots. They observed the negative effect of a back wave rising in the sprue, shown in Figure 3a, and proposed the designation of characteristic heights for this phenomenon, shown in Figure 3b. Analysis of these heights as a function of time, presented in Figure 3c, led to the conclusion that rapid height changes result in the presence of turbulences in the sprue and the whole gating system, which are almost impossible to eradicate with the application of a non-tapered sprue.

The problem of sprue tapering can be easily solved in almost every case, except for automatic molding with a horizontal parting plane. The tools and machinery used for the compaction of greensand require reverse tapering, which, in terms of the reoxidation potential, is even worse than that in previously shown cases. Possible solutions to this problem would be the application of a core with a properly tapered sprue to the mold or the machining of the sprue in the mold; however, these ideas would be extremely hard to implement. However, if performed correctly, they would decrease the rate of reoxidation caused by entrained air, increase the yield, and improve the flow kinetics. The authors in [21] tried to partially solve this problem with the addition of a well or a choke at the bottom of the sprue. Their results proved that it is possible to decrease the free surface area of the metal and reduce air entrapment. Although the amount of air entrained is high, the selected results of their optimization are presented in Figure 4. The weight of the problem is clearly visible, and attention should be paid to this topic, as castings manufactured with this method constitute a major share of total casting production.

An interesting solution is proposed in [22] and is shown in Figure 5. According to this idea, the inclination of the sprue will increase the friction between liquid metal and the mold, allowing one to obtain a decrease in velocity at the bottom of the sprue. Unfortunately, this smart idea was implemented with a lack of sprue tapering, which resulted in the increased free surface of the metal during the flow, which promoted reoxidation. It can be seen that this solution should be avoided.

Interestingly, this design was improved by the authors in [23,24] by adding tapering, and it is visible that after the initial portion of the metal passes, there is no free surface present. They also proposed a novel conical helix sprue design: at first glance, it may seem impossible to manufacture in foundry conditions; however, it is possible to assemble with a 3D-printed mold or just a 3D-printed core. The application of this type of sprue would result in an even further increase in the friction between the metal and the mold, allowing for a much-desired velocity reduction. The analysis presented in Figure 6 shows that the proposed solutions allow one to decrease the amount of air entrained during the flow.

However, the question of what a proper tapering angle is should be asked. It is obvious that gentle pressurization of the metal in the sprue is required to obtain a naturally pressurized system. The easiest way to obtain it is to adequately taper the sprue in a way that corresponds to the shape of the free-falling stream, which is hyperbolic. To design such a sprue, it can be divided into sections; for each section, the stream continuity law must be met, meaning that the product of the surface area and velocity of the metal should be constant. It should be kept in mind that too high a level of taper angle would only needlessly increase the velocity of the stream. Without its amplification velocity, in the case of gravity, casting exceeds the critical values, and an excess taper should be avoided. The perfect example of recommended and unacceptable sprue designs is shown in [25] and in Figure 7.

In [26,27,28], the authors presented studies on a gating system using hyperbolically tapered sprues. In both cases, there was no free metal surface in the sprue after the first portion of the metal passed, which can be seen in Figure 8, and there were neither bubbles nor cavitation, as is characteristic of conventional systems. These results are consistent with the results presented in [1], in [14,29], and in [30,31,32], which proves that there is a better way to design a sprue that allows for the elimination of undesirable turbulence and entrainment. Such analyses have been carried out not only for typical gravity castings but also for high-pressure die casting [33], investment casting [34], and the lost foam process [35] with and without filters in the gating system [36]. In some cases, the geometry of the gating system, mostly the sprue, can significantly affect the processes occurring in the casting mold, for example, ductile cast iron casting manufacturing using the in-mold nodularization method [37].

Metallurgical and technological development in the field of material science and foundry engineering is continuously progressing. However, every foundryman must realize that only a properly designed and manufactured gating system can deliver the metal to the mold without the deterioration of its properties. It should be kept in mind that, too often, in both industrial and scientific applications, manufacturing technologies of samples or castings are, to put it mildly, not optimal, as was shown in the quoted case studies. The authors are convinced that to truly maximize the potential of the proposed innovative advances in material engineering, it is necessary to solve the basic, often overlooked, problems associated with the design of the gating system. Therefore, such experiments were carried out, and their results are presented below.

## 2. Initial Studies of Sprue Geometry

### 2.1. Materials and Methods for Initial Studies of Sprue Geometry

The research began with computer simulations of metal flow in selected sprue designs with a thorough analysis of the results. In the initial studies, five sprue designs were examined. The analysis was performed using coarse mesh to assess the designs examined and plan the detailed analysis. In each, the base surface of the sprue was prolonged with a bend, a straight segment, and another bend connected to a mold cavity, which was a cube with base dimensions of 75 × 75 mm and a height of 150 mm. The mold cavity was not shown, as it drew attention away from important phenomena that take place in the sprue. The sprue designs analyzed in the initial studies are briefly described in Table 1. The initial surface was treated as the inlet, and the base surface was treated as the outlet.

The initial area of the sprue is considered to be completely filled with liquid metal in all cases except the first. This condition is rarely met in practice unless one of two conditions is met: (1) The entrance is filled by an offset step basin with a stopper sealing the sprue entrance. The stopper is raised, allowing the filling of the sprue only when the basin is filled to the design fill level. (2) The sprue is filled by direct contact with the nozzle of a bottom-pouring ladle, a technique known as ‘contact pouring’. The material used in the computer simulation was steel grade according to DIN 1681. The nominal composition of this steel is <0.30% C; 0.30–0.60% Si; 0.20–0.50% Mn; <0.04 % P; and <0.04% S. The minimum mechanical properties of the steel are YS = 260 MPa, UTS = 520 MPa, E = 18%, and impact strength ISOV = 22 J. The parameters of the initial sprue studies performed in MAGMASOFT (ver. 5.0) are shown in Table 2.

### 2.2. Results and Discussion for Initial Studies of Sprue Geometry

The first design (S_i_1) is a straight, non-tapered sprue that can be commonly found in foundries utilizing gating systems manufactured from refractory tubing. This design is equipped with a conical pouring basin; however, it did not influence the results, as the simulation software does not simulate the air entrained into the system by this pump-like solution. The flow analysis presented in Figure 9 shows that the falling stream is increasing in velocity due to gravity, and due to surface tension, this increase in velocity is accompanied by a reduction in surface area according to the stream continuity law. The stream is observed to be detached from the walls, forming a large area of free surface prone to oxidation. When the stream reaches the first bend, the direction change reduces velocity and causes turbulence, because the base of the sprue fills and creates a flat horizontal free surface, which the authors refer to as the ‘sprue lake’, which is visible from 260 ms. The flowing stream collides with the sprue lake, creating multiple splashes, causing the formation of cold shots. The stream flowing through the tube would create an underpressure in the space surrounding it, which would cause gases to be sucked from the mold into the sprue in a similar way to a shower curtain being attracted to the water stream. In the straight segment between two bends, the first portion of metal leaving the bend does not cover the whole surface area of the channel; in fact, the first portion covers only a small portion of the bottom, creating another large free surface. After a short time, the straight segment is filled due to the backpressure of the metal present in the mold. This creates a situation where the horizontal flat free surface present in the sprue will rise in a similar way to metal rising in the mold; the higher the mold, the longer the rise will take. The rising level is visible between 1000 and 1940 ms. When the mold is filled, a rapid growth of pressure in the system occurs, and a back wave travels from the currently filled level in the sprue to its initial area. In the presented design, the velocity of the back wave exceeds 8 m/s: this would truly create a potentially dangerous situation in the foundry, as the traveling back wave would create a massive splash (eruption) from the pouring basin, endangering all foundrymen standing nearby. The presence of turbulence allows a velocity reduction to be obtained; however, this effect has the terrible price of entraining massive amounts of oxides into the mold cavity.

The second design (S_i_2) is an evolution of the first. Associating the increase in the velocity of the free-falling stream with the stream continuity law allows us to state the optimal surface area of the bottom of the sprue, which is 180 mm^2^, which translates to Ø 15.1 mm pipe. The tapering of the initial entrance diameter to the bottom diameter is carried out with a straight line, which forms a sprue shaped like a truncated cone. The flow through the second sprue is shown in Figure 10. The stream is still partially detached from the mold; however, compared to the first sprue, it is greatly reduced. The volume of the first sprue (S_i_1) is 166.5 cm^3^, and for the second sprue (S_i_2), it is 128.3 cm^3^; the straight tapering of the sprue allows its volume to be reduced by 23%, which is a significant value. After reaching the first bend, the sprue lake forms and travels up the sprue between 220 and 400 ms. The formation of the wave is not the result of the complete filling of the mold cavity as in the first example; in this system, faster pressurization is possible because there is less void space in the sprue. The wave is smaller than in the first sprue, but unfortunately, it also exceeds 8 m/s; therefore, it would also cause splashes. After 1300 ms, the flow stabilizes, and the velocity field is almost stationary. The simple straight tapering of the sprue allows one to shorten the time of the sprue lake presence, reduce the underpressure in the sprue, and increase the yield. Due to the pattern used to manufacture this component of the gating system, there is a possibility of opting out of purchasing refractory tubing.

Further improvement was achieved by changing the tapering from straight to hyperbolic in the third design (S_i_3). Calculating such a tapering is simple; a constant product of velocity and surface area from the stream continuity law must be applied to sections of the whole sprue. The better the discretization performed, the better the fit of the sprue. An experienced pattern maker can easily fabricate such a sprue on a turning machine. The hyperbolic tapering allows a reduction in volume from 128.3 cm^3^ in (S_i_2) to an impressive 116.2 cm^3^ or 9%. This simple calculation shows that the straight-tapered sprues are an improvement over the non-tapered one; however, to obtain the best possible quality, a hyperbolic taper should be applied. Figure 11 presents the flow through such a sprue; it can be seen that the flowing metal is in constant contact with the mold, and a free surface is present only at the advancing metal front. After reaching the first bend, there is no back wave; only a slight increase in pressure in the system is visible from 240 to 600 ms. After the 600 ms velocity field is stationary, the animation of the metal flowing through this type of sprue is a still picture with only the time value changing. This shows great potential for naturally pressurized systems. The lack of an excessive free surface, the absence of turbulence, and the fast stabilization are their crucial advantages. A slight reduction in velocity is visible as a result of the increased friction between the flowing metal and the mold. The utilization of coarse mesh should be mentioned here again; in the analysis of such designs where the mesh size is large compared to the dimensional changes resulting from hyperbolic tapering, the results may be inaccurate. However, they should give even more pessimistic variants, as sharp edges may promote cavitation or turbulence.

The fourth design (S_i_4) is a possible answer to the problem presented previously concerning the connection of a circular sprue to a rectangular runner in a solution that allows the use of contact pouring. If the system were to be fed from an offset pouring basin, the best answer would be to use a rectangular sprue, which can be easily connected to a rectangular runner. However, the majority of bottom-pour ladles have round-shaped nozzles, which need to match the shape of the sprue, being the entrance to the gating system. The application of the sophisticated transitional shape of the sprue presented in Figure 12 is quite different from the commonly used junctions of a round sprue to a rectangular runner. The direct junction of the round sprue with the rectangular runner works poorly; flowing liquid metal ricochets back and forth down the runner, never properly filling the runner. This design may seem difficult to mold; however, it is possible and relatively easy in vertically parted molds or with the use of a core. The gradual transition of the shape from circular to rectangular is linked to decreasing the thickness of the sprue in one dimension and increasing it in the perpendicular dimension; of course, in order to avoid free surface formation, hyperbolic tapering is recommended. In this case, hyperbolic tapering was utilized with surface area manipulation, as the application of true hyperbolic tapering was not feasible due to the required shape of the sprue base. It is worth highlighting that this design (S_i_4) is very similar to the one previously presented (S_i_3). It can be noticed that, again, in this case, no free surface is formed except on the head of the advancing metal front. The increase in the surface area of the sprue base results in increased friction between the flowing stream and the mold, resulting in the positive effect of a reduction in velocity. The application of a slim rectangular bend results in an improved velocity distribution during the change in the flow direction. After 501 ms, the flow completely stabilizes; the velocity field is stationary until the end of the filling.

The authors believe that it is vital to underline the importance of the proper tapering of transitional sprues. It is recommended to translate the circular shape into the rectangular one as smoothly as possible, utilizing the entire length of the sprue. Figure 13 shows flow kinetics in an improperly tapered transitional sprue (S_i_5). At 160 ms, visible detachment of the metal from the mold is present. In a similar way to the first (S_i_1) and second (S_i_2) designs, a sprue lake is formed; this shape makes the back wave less dangerous, as it is minimal and is followed by pressure stabilization. However, it does not mean that this should be accepted. After 700 ms, the flow stabilizes, and again, the velocity field is stationary. Interestingly, in this design, the tapering is also conducted according to the stream continuity law, as in the previous (S_i_4) design; however, the shape transition does not take place over the whole sprue and starts after one-third of its length. This type of sprue is rarely seen; however, it should be a valuable indication for those who intend to use this solution. In Figure 14, the authors present the result of the simulation performed in NovaFlow&Solid (ver. CV 4.4r2) software, showing the trials focused on determining the optimal transition of shapes and the optimal solution used in the S_i_4 design. It can be seen that the stream falls into a large sprue lake, which would work exactly like a pump; even though contact pouring may be applied, entrainment would take place, and the casting quality would deteriorate. The authors do not present the details related to the optimal solution, as they are considered to be know-how; however, it can be said that the designs resulting from various automatic transitions are not satisfactory, and it is necessary to discretize the sprue to short sections and perform a number of transitions between them.

Initial studies showed that the lack of or improper tapering and transition cause depressurization of the sprue and promote the formation of turbulence. The stream of metal free-falls down the early part of the sprue and so does not have the advantage of boundary friction. Keep in mind that the real situation is more damaging, as the vast majority of simulation software does not simulate the presence of bubbles. The plunging stream conditions in the sprue entrain massive volumes of air, probably in a mix of at least 50/50 air/metal. This air influx will significantly deteriorate the quality of the castings by forming bifilms and bubbles. In Figure 15, the characteristic points of three selected designs are brought together to schematically show the phenomena taking place in (a) non-tapered, (b) straight-tapered, and (c) hyperbolically tapered sprues. For the non-tapered sprue, the unfilled sprue is visible alongside a sprue lake and numerous splashes and cold shots. The straight taper reduces the gaps in the initial phase of the flow; however, a metal back wave still forms. Application of the hyperbolic taper allows the eradication of voids in the sprue as well as the back wave; this design is recommended by the authors. Transitional sprues are a possibility to perform contact pouring; however, in the authors’ opinion, it may be easier to use a rectangular sprue, which would require neither vertical mold parting nor core use. However, there is potential for this design, for example, in the case of vertically parted molding lines. Under such conditions, the application of transitional sprues would allow for great cost reduction, since no pouring basin would have to be used, and state-of-the-art pouring machines are capable of supporting this process. Additionally, the volume of the sprue is minimized compared to the standard straight taper, and the quality of the casting should be enhanced as a result of the minimization of entrainment.

## 3. Studies of the ‘Short’ Sprue

### 3.1. Materials and Methods for Studies of the ‘Short’ Sprue

In a detailed analysis of short sprues, nine designs were examined. Analysis was performed using refined mesh to precisely determine the flow kinetics in the tested design solutions and conditions. In each sprue, the base surface of the sprue was extended with a bend, a straight segment, and another curved segment. In MAGMASOFT simulations, the second bend was connected to a mold cavity, which was a cube with base dimensions of 50 × 50 mm and a 400 mm height. Like in the initial studies, the mold cavity was not shown, as it drew attention away from important phenomena taking place in the sprue, and the issues related to mold filling are presented later. The sprue designs analyzed in detailed studies are briefly described in Table 3. The initial area of the sprue is considered to be completely filled with liquid metal in all cases. Again, this condition would require an offset step basin with a stopper that seals the entrance of the sprue or the application of ‘contact pouring’.

The examination consisted of flow analysis in selected short sprue designs with qualitative and quantitative analysis of the results. The simulation material was EN-GJS-500-7 ductile iron according to PN-EN-1563:2018-100. The schematic representation of the conditions tested is presented in Figure 16, and the parameters of short sprue studies performed in MAGMASOFT (ver. 5.4) are shown in Table 4.

### 3.2. Results and Discussion for Studies of the ‘Short’ Sprue

The analysis allowed the assessment of the solutions examined using a number of criteria, including air entrapment, air evacuation, flow tracer, temperature, and velocity. However, due to the aim of the work, the authors decided to present selected characteristic stages of the flow in each solution examined using the air evacuation criterion. This criterion is a valuable tool, showing how the air initially present in the system evacuates the system due to the flow of liquid metal. The figures presented below have no scale, because the analysis only requires the distinction of two colors, light and dark gray. Dark gray is the air, and light gray is the metal. Below each set of figures presenting flow kinetics, there is a graph showing the mass flow through the visible ending of each solution, which may be treated as an ingate. The first solution (S_s_1) is a straight round sprue with no tapering; Figure 17 presents the flow kinetics in this variant. It is visible that, from the beginning, the stream does not fill the whole cross-section of the sprue. Falling metal leaves a vein of air visible on the left side of the sprue. Then, the stream reaches the bends and flows at the bottom of the channel, leaving the top part unpressurized, and the metal is injected into the mold cavity with only a part of the ‘ingate’ cross-section, resulting in an elevated metal velocity. The backpressure of the metal building inside the mold causes gradual pressurization of the horizontal channel and the sprue; changes in air pocket morphology can be observed as large portions of air turn into numerous smaller bubbles traveling up the sprue. The formation of a metal back wave traveling up the sprue is visible between 249.5 and 407.4 ms, as the current position of the sprue lake, being the front of the back wave, is close to the end of the air pocket in the sprue. After 407.4 ms, the runner is pressurized, except for the underpressure void after the second bend created due to cavitation. At 471.3 ms, the sprue completely fills. The cavitation area formed after the interior radius of the bends is an undesirable phenomenon; the negative effect can be reduced with an increase in the bend radius. The maximum velocity of the metal is close to the free-fall velocity (3.71 m/s), as it exceeds 3.5 m/s.

Figure 18 presents the mass flow through the ingate of the examined system, which, in this case, would be a channel after the second bend. At around 250 ms, the flow through the ingate starts and gradually increases until 300 ms, when backpressure in the mold starts to form. This phenomenon causes the start of sprue pressurization, which results in the fluctuation of the mass flow values until 1200 ms, where the flow stabilizes and the mass flow runs almost according to a linear function. Significant fluctuations are connected to turbulence inside the system, where the lack of metal and internal friction greatly affects the mass flow. The last major fluctuation is the result of the eradication of the volume of cavitation. The most significant effect and dangerous turbulence occur from 300 to 500 ms and between 1100 and 1200 ms. The filling time is 1.9 s, which translates into an average mass flow of 2.95 kg/s, and the maximum mass flow reaches 3.7 kg/s. Calculation in MAGMASOFT allowed us to designate the amount of air to be 0.111 dm^3^, the free surface area during the flow to be 3824.73 cm^2^, and the average velocity at the ingate to be 2.87 m/s. It should be mentioned that the graph presents the mass flow function in the cross-section of the ingate, and in the sprue, changes in mass flow may be greater. In the opinion of the authors, the most important variable that should be analyzed in the angle of mass flow diagrams is the inclination of the flow after it starts to reach the bend of the function. Values of the angle closer to 90° would indicate that filling the runner/ingate is faster and that the initial portion of the metal covers a higher surface area of the channel. This value is called the ‘inclination angle’. In the case of the first design, the inclination angle is 96° for the first peak at 300 ms and 104° for the second peak at 450 ms at the maximum flow value.

The second solution (S_s_2) is a round sprue with a negative tapering angle with a choke at the bottom that is similar to the one presented on the right side in Figure 4. This solution is characteristic of automatic molding lines with a horizontal parting plane, where the use of a sprue with conventional tapering would generate problems with molding sand compaction in zones adjacent to the sprue. The presence of the choke is supposed to solve problems connected to excess volume at the bottom of the sprue and results in the proper pressurization of the system after this part. The shape of the sprue presented in this solution was suggested by automatic molding line technologists; the suggestion was to decrease the initial diameter to obtain a runner with a diameter equal to that in designs with a positive tapering angle. Analysis of Figure 19 presenting the flow kinetics in this system allows one to state that the volume of air in the sprue in the initial filling phase is greater in comparison to the straight sprue (S_s_1), and multiple veins of air are formed around the stream. When the stream reaches the choke, a large number of bubbles are formed, which is visible between 242.9 and 278.1 ms. This issue is connected to a rapid change in the stream direction as well as the formation of turbulence. After 278.1 ms, the back wave traveling up the sprue can be noticed. Pressurization of the runner, except for the cavitation void near the second bend, takes place after 418.8 ms; in contrast to the previous design, it occurs before the pressurization of the sprue. The total pressurization of the sprue is obtained at 492.4 ms. Similarly to the previous design, there is cavitation near the interior radius of the second bend, as well as at the junction of the sprue bottom with the choke and the horizontal part, which would cause oxidation and promote entrainment. Pressurization of the horizontal part of the system is improved in comparison to the first variant because of the choke, which forces the metal into the full cross-section. However, the authors would not recommend this solution. It is vital to notice that almost no simulation software is able to simulate the presence of small bubbles and bifilms, which, in this case, would be formed. The improved pressurization of the runner would come at the cost of increased metal mixing near the choke. Again, because there is no pressurization, the maximum velocity of the metal is close to free-fall velocity (3.71 m/s), as it exceeds 3.5 m/s.

Figure 20 shows the mass flow through the ingate. At around 420 ms, the flow through the ingate starts and gradually increases to 500 ms, where backpressure in the mold and sprue starts to form. Like in the previous solution, this phenomenon starts the pressurization of the sprue, which causes fluctuation at around 1000 ms. Although after 1000 ms, the system may seem stabilized, numerous minor fluctuations are present between 2000 and 4000 ms. They are considered to be the result of pressure changes in the system linked to the backpressure of the metal in the mold. The filling time is 5.1 s, which translates into an average mass flow of 0.99 kg/s, and the maximum mass flow reaches 1.3 kg/s. Calculation in MAGMASOFT allowed us to designate the amount of air to be 0.22 dm^3^, the free surface area during flow to be 2546.35 cm^2^, and the average velocity at the ingate to be 1.9 m/s. This translates into the fact that expansion with the application of the choke doubles the amount of entrapped air. This is a tremendous increase that shows the danger behind this solution. A reduction in free surface area is obtained with the presence of the choke, which prevents the flow of the metal only on the bottom part of the runner/ingate as a result of centrifugal force, as is the case in the previous design. The reduction in velocity is connected to the increased area of contact between the metal and the mold due to the presence of the choke, as well as the turbulence that takes place near it. Although a reduction in velocity is vital and desirable, it should not be obtained in this way. In the second design, the inclination angle is between 94 and 95°.

The flow kinetics for the third solution (S_s_3), which is a round sprue with a reverse taper without a choke, is shown in Figure 21. This design is also quite common in automatic molding lines with a horizontal parting plane. The initial fill phase before the stream reaches the first bend is similar to the previous design with a choke (S_s_2); the presence of the choke is the only difference between these designs. After 308.8 ms, the formation of a sprue lake and back wave can be seen with air pockets that form veins. The pressurization of the runner is analogous to that of the first design (S_s_1); the centrifugal force prevents the metal from flowing through the entire cross-section of the channel, creating a large free surface. After 482 ms, the runner is completely pressurized, whereas the sprue is pressurized at 689 ms. In this design, the maximum velocity of metal is close to free-fall velocity (3.71 m/s), as it exceeds 3.5 m/s.

In Figure 22, the mass flow is presented through the vertical part of the examined system after the second bend. At around 330 ms, the flow through the ingate starts and gradually increases to 350 ms, where backpressure in the mold and at the bottom of the sprue starts to form. Like in the previous solution, this phenomenon starts the pressurization of the sprue, which causes fluctuation between 350 and 400 ms due to air evacuation. At about 1000 ms, a significant fluctuation occurs again as a result of cavitation volume eradication. After pressurization and relative stabilization of the flow, three visible fluctuations can be noticed at 1900 ms, 2250 ms, and 2500 ms, which are considered to be a result of pressure changes in the system linked to the backpressure of metal in the mold. The filling time is 4.1 s, which translates to an average mass flow of 1.21 kg/s, and the maximum mass flow reaches 1.6 kg/s. Calculation in MAGMASOFT allowed us to designate the amount of trapped air to be 0.251 dm^3^, the free surface area during the flow to be 2130.37 cm^2^, and the average velocity at the ingate to be 2.32 m/s. The lack of a choke results in an increase in air entrapment; however, the free surface area is reduced. This solution, as well as the previous one, traps more air than the initial one. A small reduction in velocity is achieved, which can be an effect of turbulence in the system, especially near the sprue lake; however, it again causes an increase in entrapped air. This design should be avoided. In the third design, the inclination angle is 94° for the first peak at 300 ms and 96° for the second peak. Like in the previous design, the shape of this solution was suggested by automated molding line technologists; the suggestion was to decrease the initial diameter to obtain a runner with a diameter equal to the one in designs with a positive tapering angle.

The fourth design (S_s_4) is a popular modification of the first. Foundry engineers often tend to use prolongation of the sprue to the level below the runner entry. It is said to cushion the impact and improve the flow kinetics. It does no such thing. Figure 23 shows the filling of the sprue with a well at its bottom, which is a twin design compared to the first, except for the well. The filling of the sprue is almost the same as in the first solution. However, it can be seen that after 235.9 ms, when the stream reaches the bottom, the back wave travels up the sprue much faster than in the first solution. The full pressurization of the sprue is reached at 367.1 ms. It should be underlined that when the head of the flowing stream reaches the well, a rapid change in direction is forced, which, in the presence of air pockets, forms perfect conditions for entrainment. Filling of the runner starts after 235.9 ms, and the metal jets from the sprue to the runner swirl tangentially to the diameter of the channel; after reaching the second bend, this whirlpool collapses and creates turbulence. Cavitation is also present in a way similar to that in the shown designs. When the system is pressurized and flow takes place, the well is almost a dead zone with little metal exchange with the rest of the system. Analysis of the flow tracers shows that this excess space hosts an underpressurized whirlpool that is constantly rolling during the flow, which could lead to suction of the gases from the mold and their introduction into a flowing stream of metal in contact with the well whirlpool presented in Figure 24. The maximum velocity in the system also exceeds 3.5 m/s.

The mass flow through the ingate in the examined system after the second bend is shown in Figure 25. The flow starts around 330 ms and gradually increases to 400 ms, when the backpressure in the mold starts. After pressurization from 400 ms, a series of fluctuations are present until 1100 ms and between 1300 and 1600 ms. The filling time is 2.2 s, which provides an average mass flow of 2.58 kg/s, and the maximum mass flow reaches 3.2 kg/s. Calculation in MAGMASOFT allowed us to designate the amount of air to be 0.0894 dm^3^, the free surface area during the flow to be 3057.66 cm^2^, and the average velocity at the ingate to be 2.97 m/s. The presence of the well decreases the amount of entrapped air compared to previous designs; nevertheless, it presents a higher free surface area compared to sprues with reverse tapering. The increase in free surface is believed to be related to the formation of a whirlpool-like distribution of metal in the runner. In the initial filling phase, centrifugal force induces rotating liquid metal to stick to the mold walls, leaving the center of the channel empty, and thus, the free surface is increased. The velocity is slightly increased compared to the initial design. Again, this design should be avoided, as it is not a major improvement compared to the initial one. The fourth design is characterized by an inclination angle of 95°.

The fifth design (S_s_5) is the first to utilize a positive tapering angle that minimizes the amount of air surrounding the flowing stream. The kinetics of the flow for this variant is presented in Figure 26. An improvement in comparison to the versions shown earlier is clearly visible. Straight-line tapering does not completely eradicate the air pockets; however, it manages to minimize their volume. The falling stream is not in contact with air over the entire length of the sprue, but only a part up to 70 mm from the head of the stream is surrounded by veins of air. At 197.6 ms, when the stream reaches the bend, a large bubble is formed. Complete pressurization of the sprue is achieved after 215.6 ms, and that of the runner occurs after 369.6 ms. This design also presents a cavitation void near the ingate. In contrast to the previous designs, due to the presence of the tapering, which increases the friction during the flow, the velocity of the falling stream does not exceed 3.5 m/s in the sprue; however, the maximum value is close to 3.5 m/s. However, after passing the first bend, centrifugal force compacts the stream, which for a short time exceeds 3.5 m/s. This issue is connected to decreased turbulence at the bottom of the sprue, which consumes a portion of energy, resulting in a decrease in velocity. Additionally, the tapering angle results in a decreased surface area at the bottom of the sprue, which, after pressurization in accordance with the stream continuity law, must present higher velocity. One may think that the designs shown might be better due to the possibility of velocity reduction by the presence of an increased volume. Unfortunately, this would not be the right way; it is vital to emphasize that there are relatively easy ways of delivering the stream to the mold cavity with decreased velocity that do not require its reduction through the occurrence of turbulence, which happens if there is excess volume in the system.

Figure 27 presents the mass flow through the ingate in the fifth solution after the second bend. The flow starts around 240 ms and gradually increases to 280 ms, where the backpressure in the mold generates values that resist the flow in a significant way. Following pressurization from 280 ms, a series of fluctuations occur until 1500 ms, which are associated with backpressure in the mold. After 1500 ms, the flow in the examined sprue stabilizes and gradually decreases until the end of filling. The filling time is 2.9 s, which provides an average mass flow of 1.80 kg/s, and the maximum mass flow reaches 2.7 kg/s. Calculation in MAGMASOFT allowed us to designate the amount of trapped air to be 0.0749 dm^3^, the free surface area during the flow to be 591.21 cm^2^, and the average velocity at the ingate to be 3.76 m/s. The lack of excess volume at the bottom of the sprue in the form of a well or choke, along with tapering, reduces the turbulence and smoothens the direction transition, which affects the velocity, which is higher in this solution. The amount of entrained air is decreased compared to all of the previously simulated variants, and the free surface of metal formed during the flow is half the order of magnitude lower. This presents the problem of why the application of a tapering angle is necessary. This design can be treated as permissible; however, as is shown later, optimization of the tapering angle results in a further reduction in the entrained air, free surface, and velocity, which should be the goal of the optimization process. The inclination angle for the fifth design is 94°.

The sixth design (S_s_6) is an evolution of the fifth. The initial area of the sprue and the one at the bottom are the same as in the previous solution; however, the tapering is changed from straight to hyperbolic. The flow through the sixth variant is presented in Figure 28. This potentially small change allows one to reduce the length of air pockets located near the advancing metal front. The fall of the stream fills the available area of the channel, generating a situation in which liquid metal can be treated as a piston pushing the air out of the channel. The pressurization of the sprue is completed at 234.8 ms, and that of the runner is completed at 252.7 ms. Cavitation near the bends is still present. During the initial filling phase, the fall velocity is around 3 m/s, and the decreased velocity is the result of increased friction between the stream and the mold, as the stream is in constant contact with the mold wall. However, as a similar bend effect to the one in the previous design occurs, the centrifugal force causes uneven runner filling and an increase in velocity.

The mass flow through the sixth ingate design is shown in Figure 29. The flow starts at 230 ms and quickly increases to the maximum value reached at 250 ms; such a short pressurization time is connected to almost no free surface during the flow except for the front, which translates to no air pockets and, additionally, backpressure from the mold cavity. Between 250 and 1130 ms, a number of small fluctuations appear, after which the flow stabilizes and gradually drops until the mold is filled. The filling time is 3.0 s, which provides an average mass flow of 1.73 kg/s, and the maximum mass flow reaches 2.5 kg/s. The calculation in MAGMASOFT allowed the design of the amount of trapped air to be 0.0706 dm^3^, the free surface area during the flow to be 515.72 cm^2^, and the average velocity at the ingate to be 3.13 m/s. Further minimization of excess volume along the whole length of the sprue results in a reduction in turbulence. Both the amount of air entrained and the free surface of metal formed during the flow are slightly but visibly reduced. Additionally, increased friction in a system results in a much-desired reduction in velocity. A comparison of these variants’ results with the previous one proves that there is a direct impact of the tapering course on the kinetics of the filling. In circular channels, the separation of the stream without the generation of turbulence and entrainment is almost impossible. Additionally, the tapering of a runner with a round cross-section requires some creativity from the technologist, and it is much easier to use a different shape, which is presented later. The sixth design presents an inclination angle between 93 and 94°.

The seventh design (S_s_7) incorporates a square sprue with hyperbolic tapering; the flow kinetics for this version is presented in Figure 30. However, this sprue has an identical cross-sectional area at each level, as the round sprue with hyperbolic tapering instead of round tapering is square. In this design, the ratio of the width of the bottom of the sprue to height is b/h = 1, which translates into a hydraulic diameter equal to 11.77 mm. This design is the first of three that presents an increased perimeter relative to the hyperbolic round sprue. The aim of an increase in perimeter is an increase in friction, which should result in reduced turbulence and velocity. The flow kinetics is similar to that in the previous designs until the stream reaches the first bend; however, slightly more small air pockets are present. The smaller hydraulic diameter translates into counteraction of the formation of turbulence, since there is reduced space for a large wave to form. The sprue is completely pressurized at 232.8 ms, which is an improvement over the round hyperbolic sprue. However, pressurization of the runner is achieved later on. The complete pressurization of the runner is obtained at 344.9 ms. Cavitation near the second bend is still present but slightly smaller compared to the previous designs. The head of the falling stream reaches a velocity of around 3 m/s.

Figure 31 shows the mass flow through the ingate. At around 230 ms, the flow through the ingate starts and increases until 270 ms, at which time, backpressure in the mold forms. Until 500 ms, major fluctuations in the mold are present; the flow stabilizes near 1700 ms; however, a fluctuation at around 2200 ms is visible. The filling time is 2.9 s, which translates to an average mass flow of 1.79 kg/s, and the maximum mass flow reaches 2.66 kg/s. The calculation in MAGMASOFT allowed us to designate the amount of air to be 0.065 dm^3^, the free surface area during the flow to be 511.5 cm^2^, and the average velocity at the ingate to be 3.39 m/s. Analysis of the results allows us to state that, in this design, the values of entrapped air and the free surface are lower compared to the versions shown previously. These results are the first indication that an increase in the perimeter of the hyperbolically tapered sprue may reduce the potential for reoxidation during flow. Interestingly, though the number of air pockets in the sprue during the flow seems higher than in the previous design (S_s_6), the quality of the flow in terms of generating oxidation defects is improved. In the authors’ opinion, this phenomenon is connected to better behavior and less turbulence of the stream when it flows through the bend. The rectangular cross-section allows better flow kinetics in these responsible parts of the mold. Unfortunately, the velocity value is higher than in a round hyperbolical solution. It can be again connected to the centrifugal force that forces flow, which can be seen in Figure 30 at 232.8 ms, where a free surface is created, and the metal flows only through a section of the channel’s whole cross-section, which results in an increased velocity. The inclination angle in this design is between 93 and 94°.

The eighth design (S_s_8) is an evolution of the previous designs; it uses a rectangular sprue with hyperbolic tapering. Figure 32 presents the flow kinetics of this variant. Again, this sprue has an identical cross-sectional area at each level to those of the round and square sprues with hyperbolic tapering. This design presents a width-to-height ratio equal to b/h = 1.625, which translates into a hydraulic diameter equal to 11.43 mm, slightly lower than the previous ones. Rough examination of the flow suggests that the flow is almost identical to the square sprue, and small air pockets are present. However, the reduction in the channel allowed us to obtain better flow near the bends, which is a desirable outcome. The sprue fills completely at 212.1 ms, and the runner fills completely at 321.1 ms. A cavitation area is still present. The velocity of the stream in the initial filling phase is below 3 m/s.

Analysis of the mass flow through the eighth ingate design presented in Figure 33 shows that the flow starts at 240 ms and quickly increases to the maximum value, which is reached at 250 ms. This short pressurization time is connected to the minimal free surface during the flow, except for the front of the stream, and the building of backpressure from the mold cavity. After pressurization, a series of major fluctuations occur, which are associated with the collapse of the cavitation area in the ingate. A visible fluctuation is present at around 800 ms, and two minor but visible fluctuations are present between 2000 and 2400 ms. The filling time is 3.0 s, which provides an average mass flow of 1.73 kg/s, and the maximum mass flow reaches 2.5 kg/s. Calculation in MAGMASOFT allowed us to designate the amount of trapped air to be 0.0509 dm^3^, the free surface area during the flow to be 619.11 cm^2^, and the average velocity at the ingate to be 3.07 m/s. An additional increase in perimeter allowed for the reduction in the entrained air volume as well as the velocity of the metal. It is unfortunate that the free surface area was increased, which is believed to be a result of the collapse of the cavitation zone in the initial mold-filling phase. It should be said that the analysis of the sprue did not contain solutions that allowed the eradication of the cavitation problem, as they greatly influenced the results. In this design, the inclination angle is 93°.

The ninth design (S_s_9) incorporates a thin rectangular sprue with hyperbolic tapering. The flow kinetics for this solution is shown in Figure 34. This sprue has an identical cross-sectional area at each level to those of the round, square, and rectangular sprues with hyperbolic tapering; however, it presents the highest value of the perimeter. In this design, the ratio of the width of the sprue bottom to its height is b/h = 2.89, which translates into a hydraulic diameter equal to 10.29 mm and is the slimmest variant tested. The flow kinetics is very similar to that in the round hyperbolically tapered sprue, better than in two previous rectangular designs. The liquid metal works as a piston that pushes air out of the cylinder. In the version with a rectangular cross-section, the smaller height of the channel counteracts the formation of turbulence because there is not enough height for a large wave to form. The sprue is completely pressurized at 212.7 ms, which is the shortest pressurization time for the sprue among the analyzed designs. This design perfectly meets the conditions of a naturally pressurized system, as after passing the metal front, the system is gently pressurized. The complete pressurization of the runner is obtained at 276.1 ms. Cavitation near the second bend is present; however, it is much smaller in comparison to the previously analyzed designs. The cavity also collapses much faster, which can be seen at the last stage of the flow for this variant. The first portion of the metal reaches a fall velocity of 3 m/s, which is a result of pressurization and the increased perimeter of the channel, which increases friction, thus reducing the velocity from the free-fall velocity, which was calculated to be 3.71 m/s.

The mass flow through the ingate in the last examined system after the second bend is shown in Figure 35. The flow starts around 250 ms and gradually increases until 270 ms, when the backpressure in the mold forms and the reduction in mass flow begins. The curve presenting the mass flow is the smoothest of all, which is a positive phenomenon. A sharp fluctuation is visible near 1770 ms. The filling time is 3.0 s, which provides an average mass flow of 1.73 kg/s, and the maximum mass flow reaches 2.23 kg/s. Calculation in MAGMASOFT allowed us to designate the amount of air entrapped to be 0.0442 dm^3^, the free surface area during the flow to be 429.42 cm^2^, and the average velocity at the ingate to be 2.79 m/s. The high b/h ratio indicates that the channel results in the obtainment of the lowest velocity value among the tapered sprues. Additionally, this allows for more control over turbulence formation, which results in the lowest free surface area along with the lowest entrapped air volume. The results show the potential for slim rectangular channels that, in the authors’ opinion, surpass round ones not only in the flow kinetics but also in the ease of molding and the preparation of the manufacturing technology. The last design obtains an inclination angle between 92 and 93°.

Figure 36 shows the summary of air entrapment values for each of the solutions examined. It can be clearly seen that the best results were achieved with positively tapered sprues. Straight tapering allows us to obtain a reduction in the entrained air. Further reduction is possible with the utilization of hyperbolic tapering, and the slimmer the channel, the less air entrained by the system. For the thin rectangular taper (b/h = 2.89), the amount of air entrained is more than halved compared to the initial design. However, the application of a well reduces the air entrapment compared to the initial design, but it is not nearly as good as a proper taper. Solutions utilizing a reverse taper produced the worst results, even though their surface area is reduced compared to the rest of the designs and they deliver smaller volumetric flow. This is an important sign for users of automatic molding with a horizontal parting that a change in their approach is necessary.

A summary of the smooth filling criterion that presents the free surface formed during the flow is shown in Figure 37. The highest free surface was obtained for the initial solution with no tapering. The application of a well allowed for a reduction in the surface, and a surprisingly similar effect was achieved for the sprues with reverse tapering. The presence of a choke in an expanded system resulted in an increased free surface compared to a system without it because of the fact that the examined choke caused undesirable kinetics of runner filling. All positively tapered sprues exhibited significantly smoother filling with a free surface up to an order of magnitude smaller than the initial design. Unfortunately, the flow from the sprue to the runner through the first bend resulted in the moderate performance of the thick rectangular sprue. There is a visible difference between hyperbolically tapered sprues and straight tapered ones in favor of the hyperbolic, with the exception of the thick rectangular one. There is a small difference between round and square hyperbolic sprues; however, there is a large difference between them and the thin rectangular one. The thin rectangular sprue exhibited the best performance in this criterion, exhibiting a decrease in the free surface by almost a factor of ten compared to the initial non-tapered sprue.

Figure 38 shows a summary of the average velocities in the ingate for the examined sprue solutions. The lowest values of velocity were obtained for two solutions with a reverse tapering angle, which presented the most turbulence and had to transfer a decreased volume of flow. It is interesting that the increase in perimeter, resulting in increased friction and a reduction in turbulence, allowed for the thin rectangular sprue with hyperbolic tapering to take third place. Surprisingly, the initial design was ranked behind the first three but before the design utilizing the well. This sheds a negative light on the theory that the well allows us to obtain a velocity reduction. The rest of the tapered ones were ranked behind the mentioned solutions. The least velocity reduction was obtained for the straight tapered round sprue, which did not contain the stream during the fall, and its tailored bottom resulted in the least friction with limited turbulence near it. As was mentioned, the velocity in the non-tapered design is a result of turbulence at the bottom of the sprue. There are other ways to reduce the velocity of metal entering the mold cavity, and creating turbulence is not recommended.

## 4. Conclusions

The analysis allowed the formation of the following conclusions:To avoid reoxidation defect formation, such as bubbles, bifilms, and dross, the sprue must be tapered.Short sprues can be manufactured with a straight-line taper, and long sprues will especially benefit from a hyperbolic taper.Tapered sprues entrain air only at the beginning of the filling process; sprues without tapering entrain air during the entire process.Sprue tapering allows a significant increase in yield; a 60% increase was obtained for the hyperbolical tapering.The use of rectangular sprues improves flow kinetics in bends, which allows one to decrease the entrainment value and the system to be pressurized. Additionally, due to their larger perimeter, rectangular sprues cause more friction between the flowing stream and the mold, allowing greater velocity reduction than round sprues.The proper shape of the sprue will cause the minimization of air entrainment into the metal. The maximum difference in the first three seconds of the flow was one order of magnitude, and for one minute, it rose to two orders of magnitude in favor of hyperbolical designs.The lack of or improper tapering of the sprue causes the formation of a back wave that travels up the sprue, which can be dangerous to foundrymen close to the mold.

## 5. Patents

Próbnik do zmiennoobwodowej próby lejności ciekłych stopów metali. Patent PL235863, Dojka, R.; Jezierski, J.; Jaromin, M.; Janerka, K.; Kaczkowski, W. 2 November 2020.

## Figures and Tables

**Figure 1 materials-15-04937-f001:**
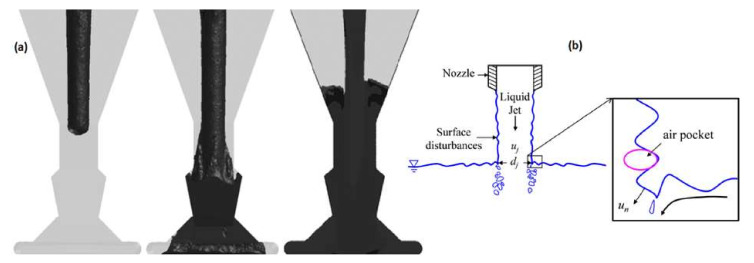
Rising metal level in improperly tapered sprue (**a**) [16]; air entrainment mechanism (**b**) [13].

**Figure 2 materials-15-04937-f002:**
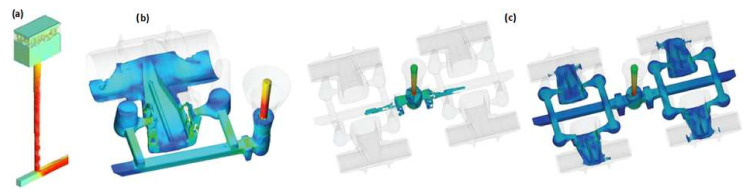
Lack of pressurization in the bottom part of sprue (**a**) [17]; unpressurized sprue (**b**) [18] and (**c**) [19].

**Figure 3 materials-15-04937-f003:**
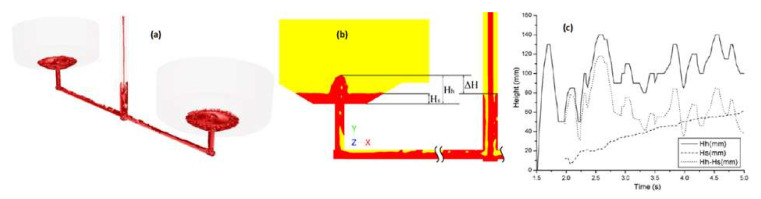
Initial mold-filling phase (**a**), schematic presentation of characteristic heights (**b**), and values of characteristic heights as a function of time (**c**) [20].

**Figure 4 materials-15-04937-f004:**
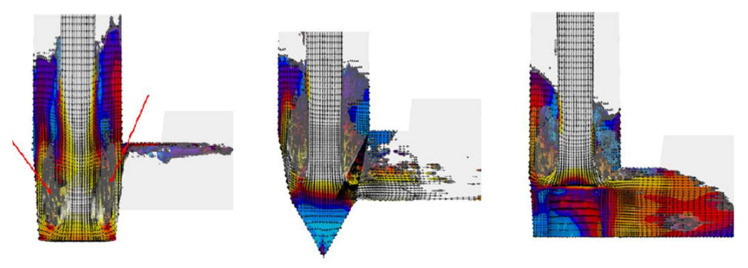
Initial filling phase of the sprue in expanded sprues (with reverse tapering) characteristic of automatic molding with horizontal parting plane [21].

**Figure 5 materials-15-04937-f005:**
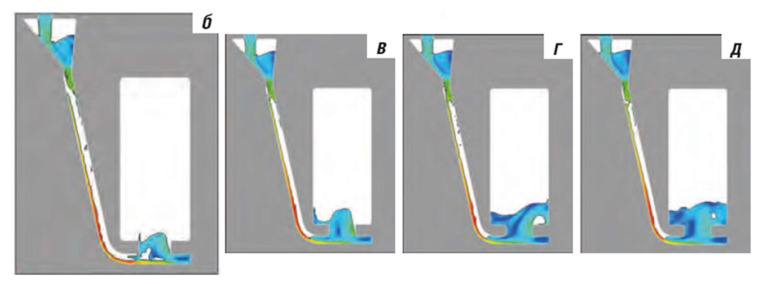
Inclined sprue without tapering [22].

**Figure 6 materials-15-04937-f006:**
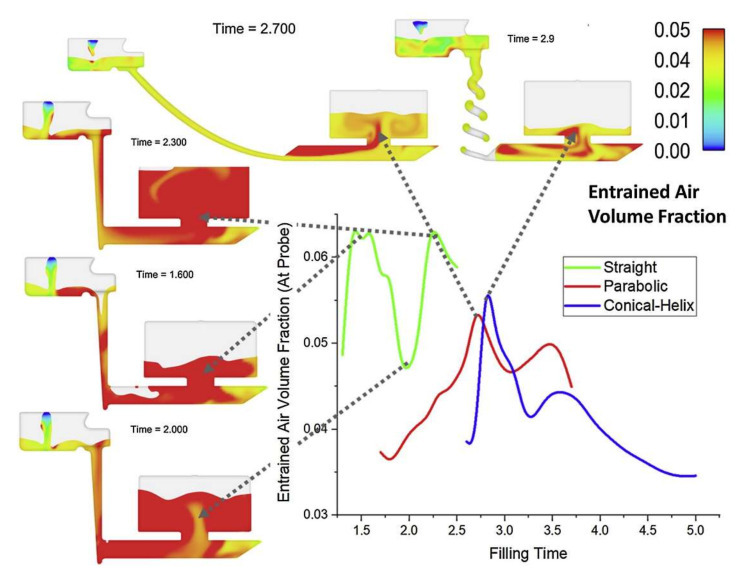
Comparison of air entrained by straight, parabolically inclined, and conical helix sprues [24].

**Figure 7 materials-15-04937-f007:**
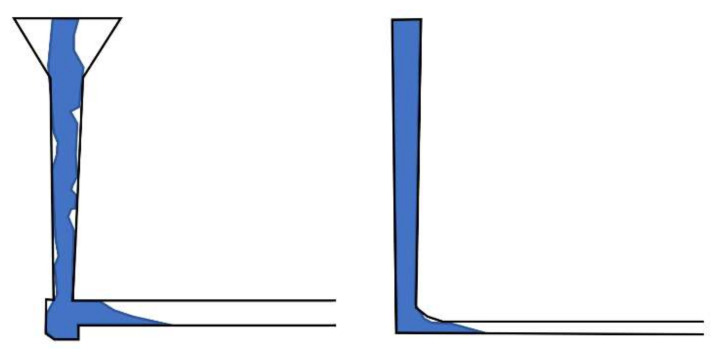
Sprue with improper (**left**) and proper (**right**) tapering [25].

**Figure 8 materials-15-04937-f008:**
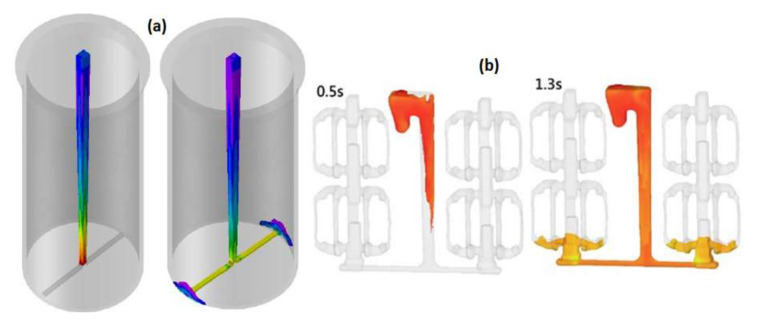
Flow in hyperbolically tapered sprues without free surface after the first portion of the liquid alloy for sleeve casting (**a**) [26] and support bracket (**b**) [27] castings.

**Figure 9 materials-15-04937-f009:**
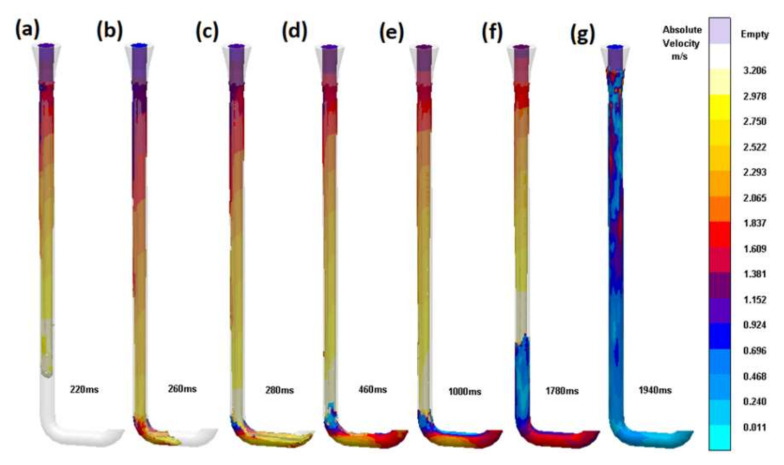
Round, non-tapered sprue (S_i_1). Subsequent stages of the flow (**a**) 220 ms, (**b**) 260 ms, (**c**) 280 ms, (**d**) 460 ms, (**e**) 1000 ms, (**f**) 1780 ms, (**g**) 1940 ms.

**Figure 10 materials-15-04937-f010:**
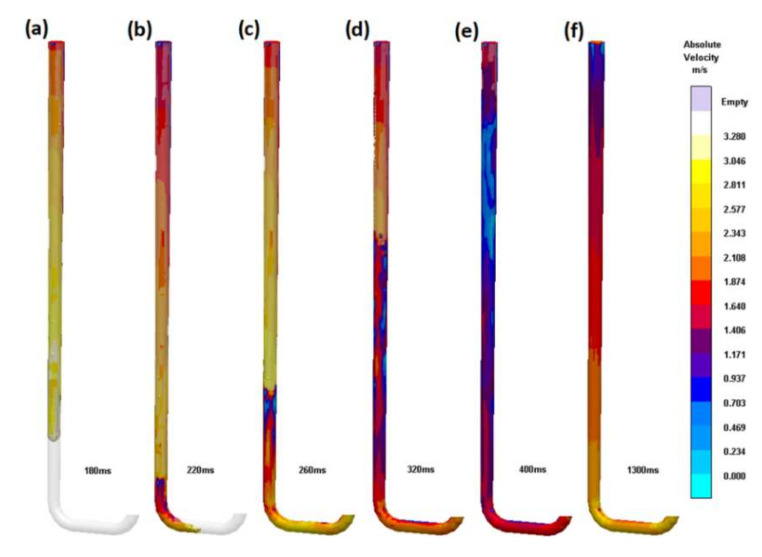
Round sprue with straight taper (S_i_2). Subsequent stages of the flow (**a**) 180 ms, (**b**) 220 ms, (**c**) 260 ms, (**d**) 320 ms, (**e**) 400 ms, (**f**) 1300 ms.

**Figure 11 materials-15-04937-f011:**
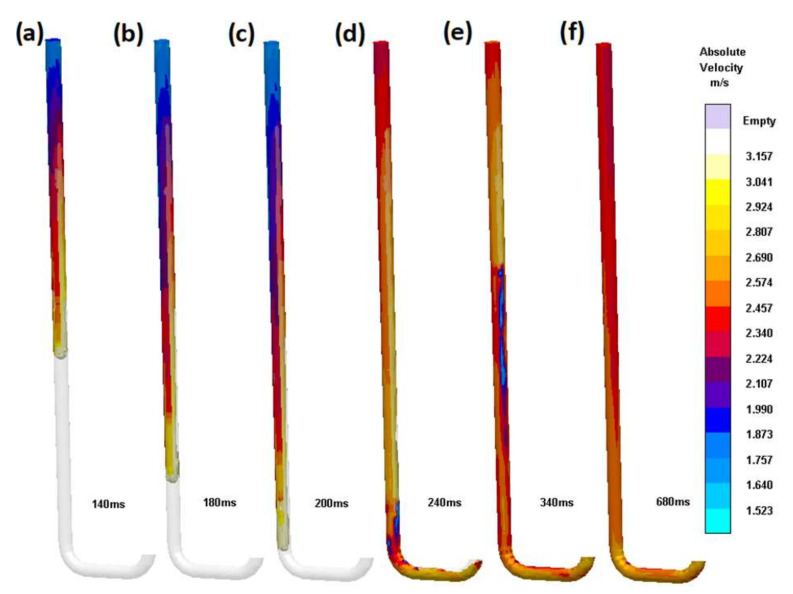
Round sprue with hyperbolic taper (S_i_3). Subsequent stages of the flow (**a**) 140 ms,(**b**) 180 ms, (**c**) 200 ms, (**d**) 240 ms, (**e**) 340 ms, (**f**) 680 ms.

**Figure 12 materials-15-04937-f012:**
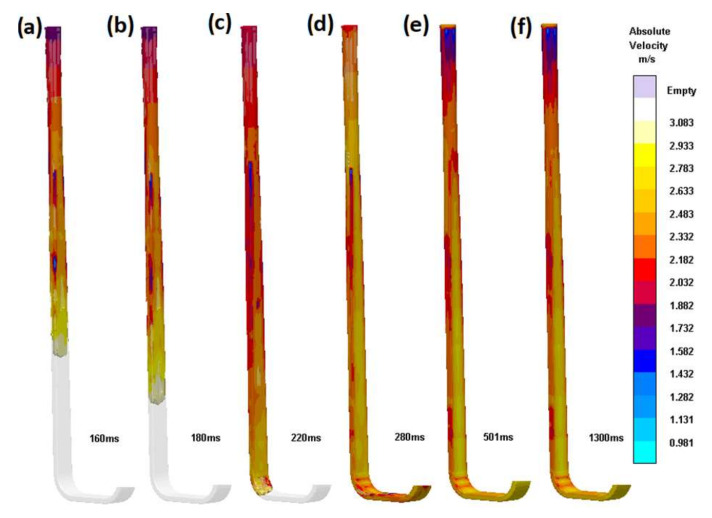
Transitional sprue with proper transition (S_i_4). Subsequent stages of the flow (**a**) 160 ms, (**b**) 180 ms, (**c**) 220 ms, (**d**) 280 ms, (**e**) 501 ms, (**f**) 1300 ms.

**Figure 13 materials-15-04937-f013:**
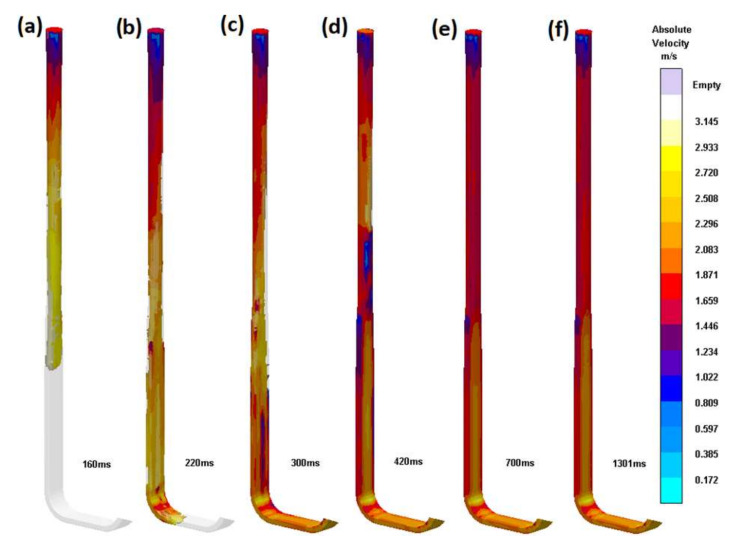
Transitional sprue with improper transition (S_i_5). Subsequent stages of the flow (**a**) 160 ms, (**b**) 220 ms, (**c**) 300 ms, (**d**) 420 ms, (**e**) 700 ms, (**f**) 1301 ms.

**Figure 14 materials-15-04937-f014:**
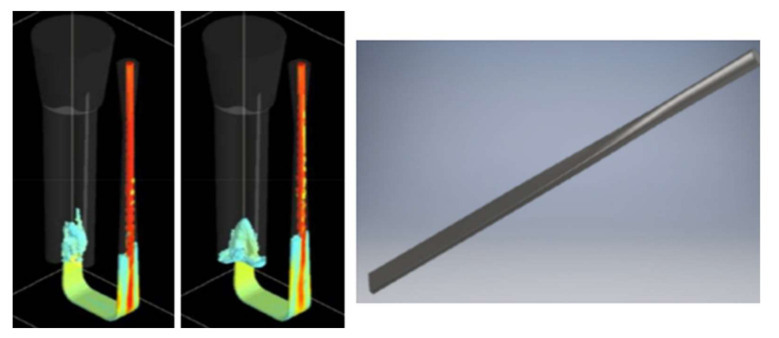
Improper tapering of the transitional sprue (**left**); 3D model of the properly tapered sprue used in initial studies (**right**).

**Figure 15 materials-15-04937-f015:**
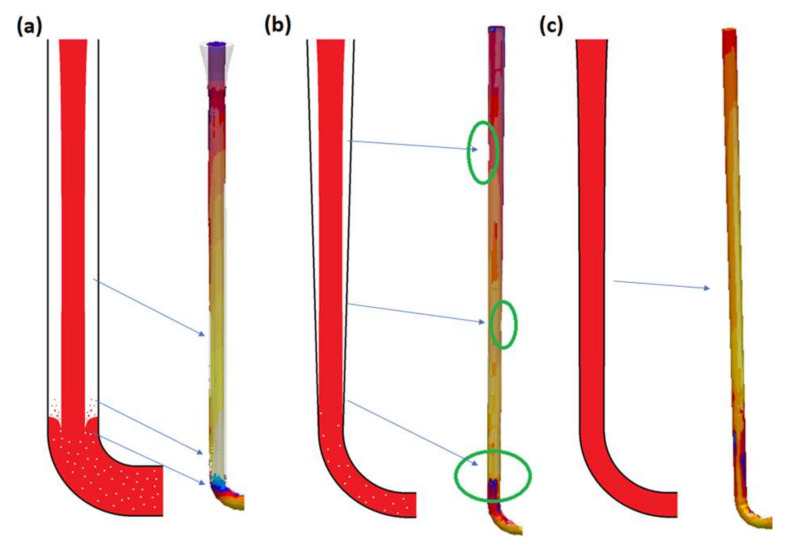
Characteristic points in non-tapered (**a**), straight tapered (**b**), and hyperbolically tapered (**c**) sprues.

**Figure 16 materials-15-04937-f016:**
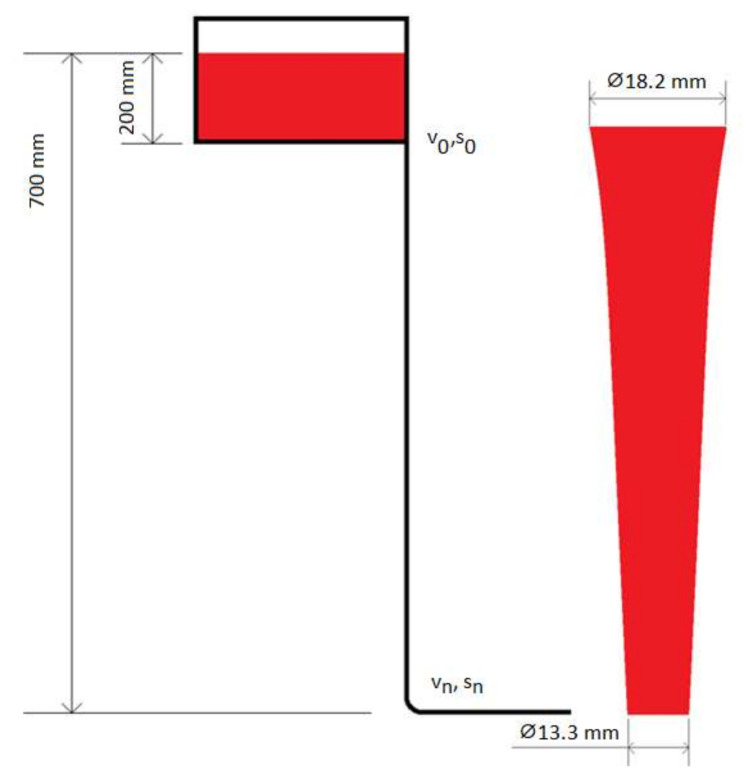
Schematic representation of the tested conditions.

**Figure 17 materials-15-04937-f017:**
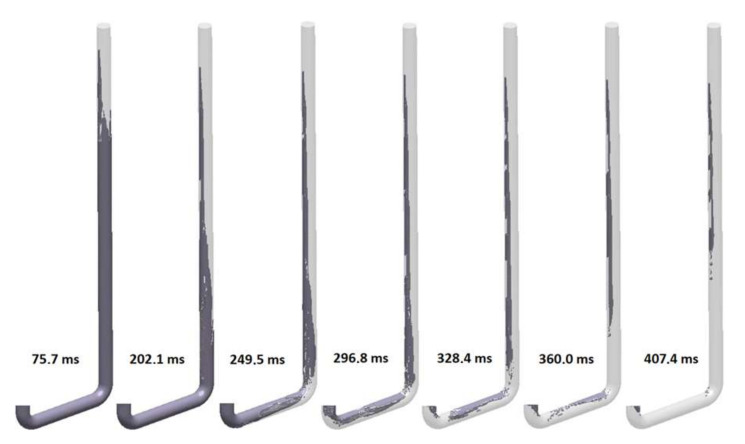
Air evacuation in round, non-tapered sprue (S_s_1), MAGMASOFT.

**Figure 18 materials-15-04937-f018:**
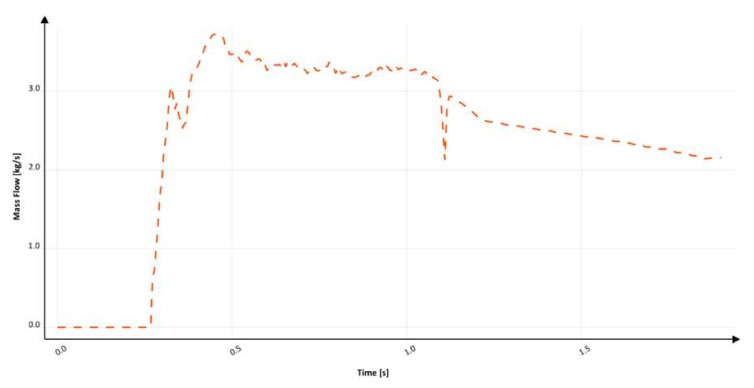
Mass flow through ingate with round, non-tapered sprue (S_s_1), MAGMASOFT.

**Figure 19 materials-15-04937-f019:**
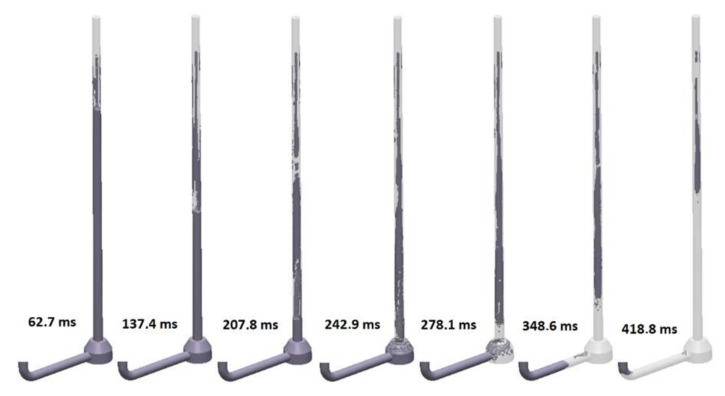
Air evacuation in round, reverse tapered sprue with choke (S_s_2), MAGMASOFT.

**Figure 20 materials-15-04937-f020:**
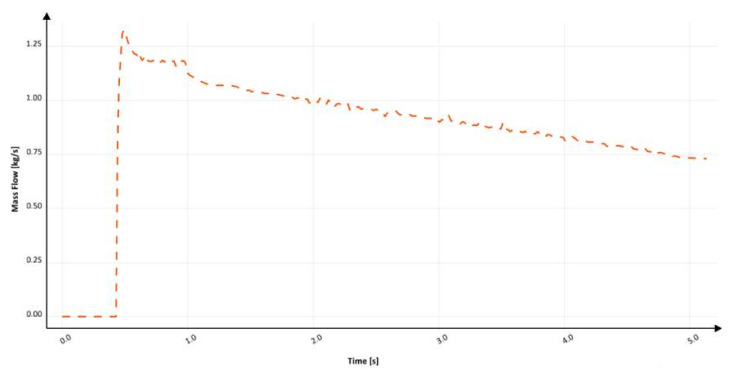
Mass flow through ingate with round, reverse tapered sprue with choke (Ss2), MAGMASOFT.

**Figure 21 materials-15-04937-f021:**
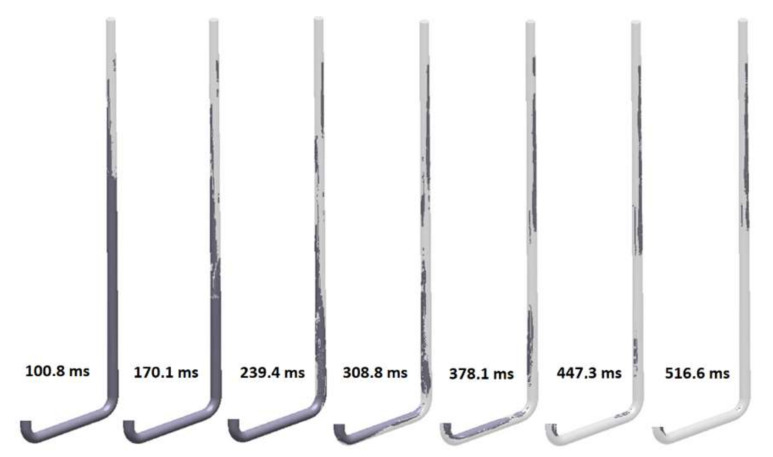
Air evacuation in round, reverse tapered sprue (S_s_3), MAGMASOFT.

**Figure 22 materials-15-04937-f022:**
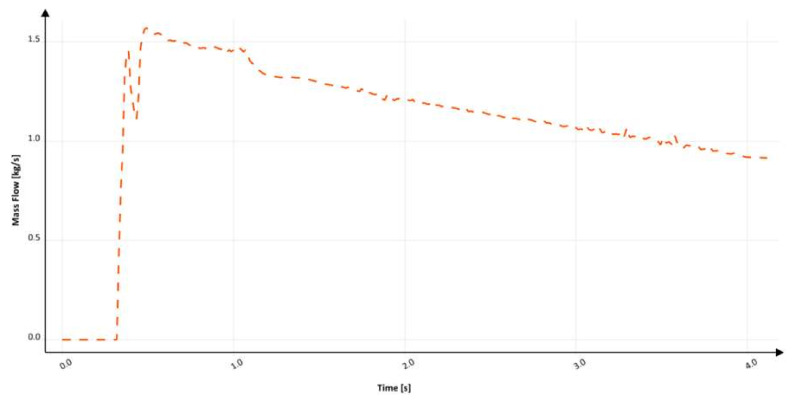
Mass flow through ingate with round, reverse tapered sprue (S_s_3), MAGMASOFT.

**Figure 23 materials-15-04937-f023:**
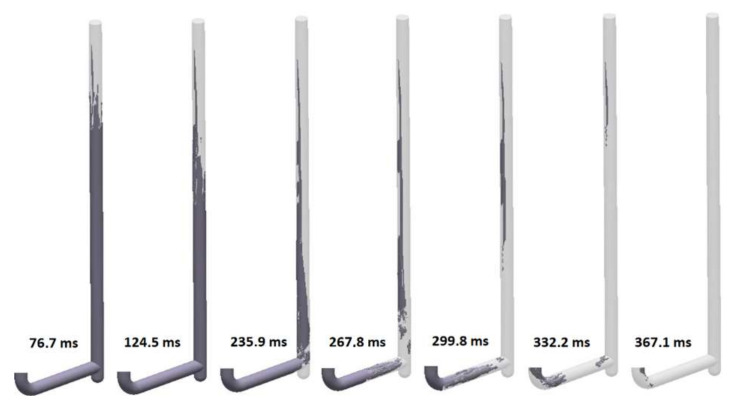
Air evacuation in round, non-tapered sprue with well (S_s_4), MAGMASOFT.

**Figure 24 materials-15-04937-f024:**
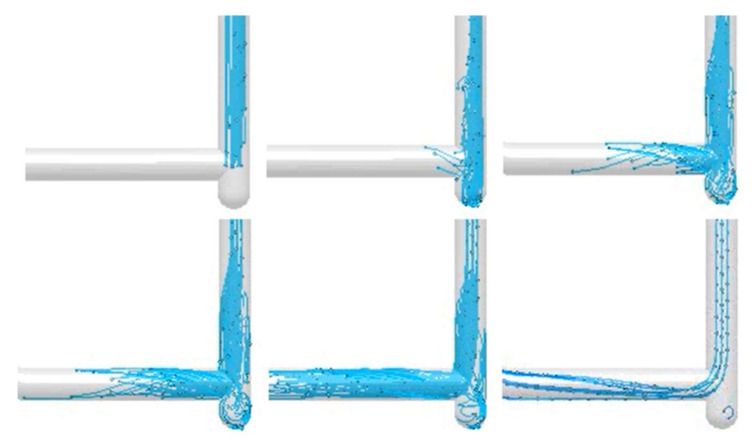
Flow tracer in the sprue well (S_s_4), MAGMASOFT.

**Figure 25 materials-15-04937-f025:**
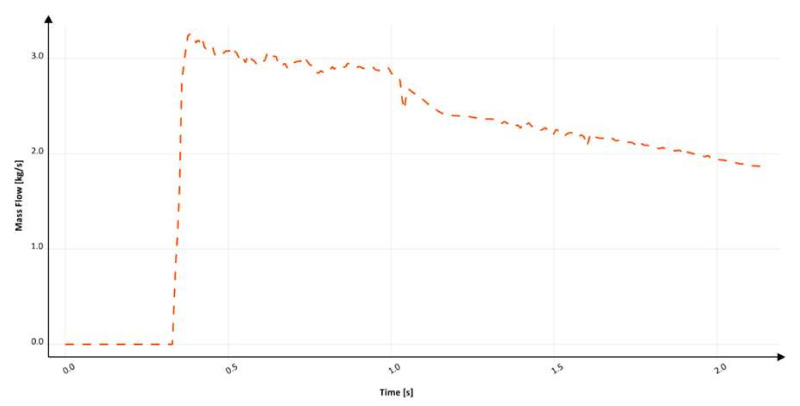
Mass flow through ingate with round, non-tapered with well (S_s_4), MAGMASOFT.

**Figure 26 materials-15-04937-f026:**
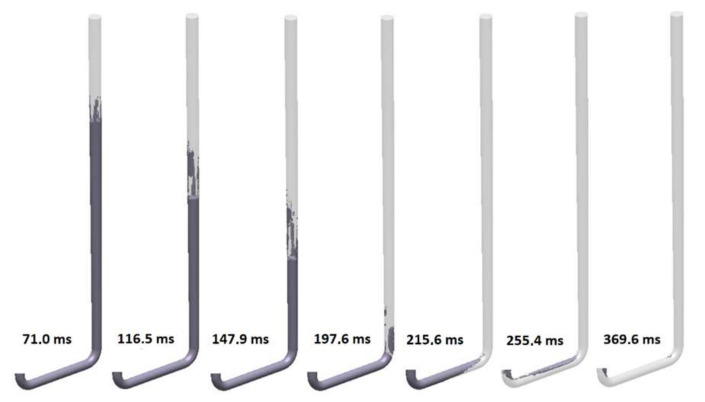
Air evacuation in round sprue with straight taper (S_s_5), MAGMASOFT.

**Figure 27 materials-15-04937-f027:**
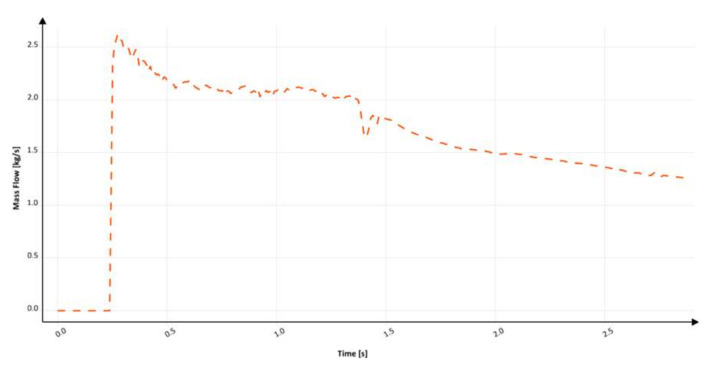
Mass flow through ingate with round sprue with straight taper (S_s_5), MAGMASOFT.

**Figure 28 materials-15-04937-f028:**
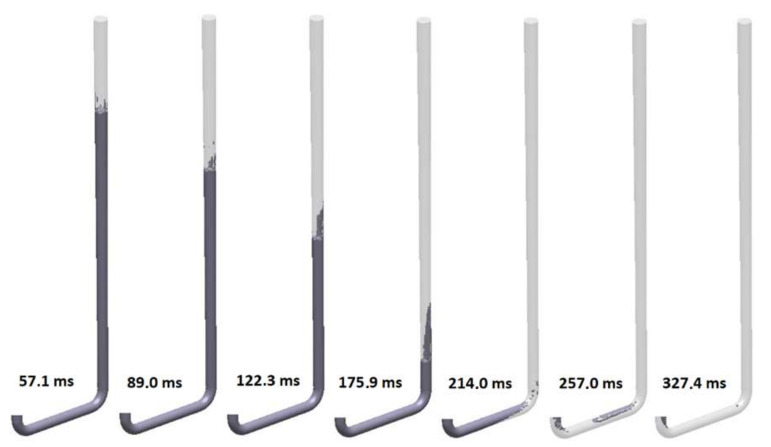
Air evacuation in round sprue with hyperbolic taper (S_s_6), MAGMASOFT.

**Figure 29 materials-15-04937-f029:**
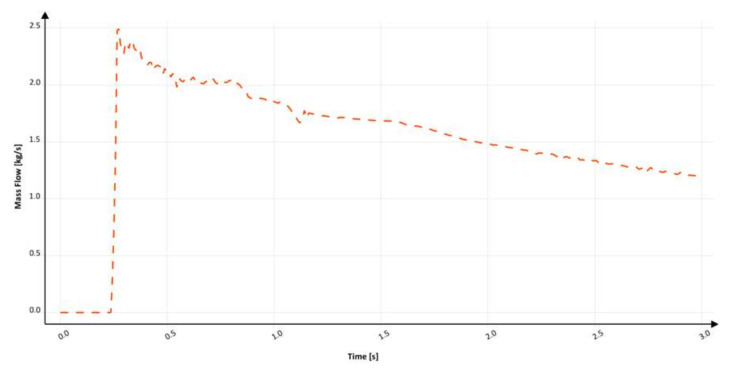
Mass flow through an ingate with round sprue with hyperbolic taper (S_s_6), MAGMASOFT.

**Figure 30 materials-15-04937-f030:**
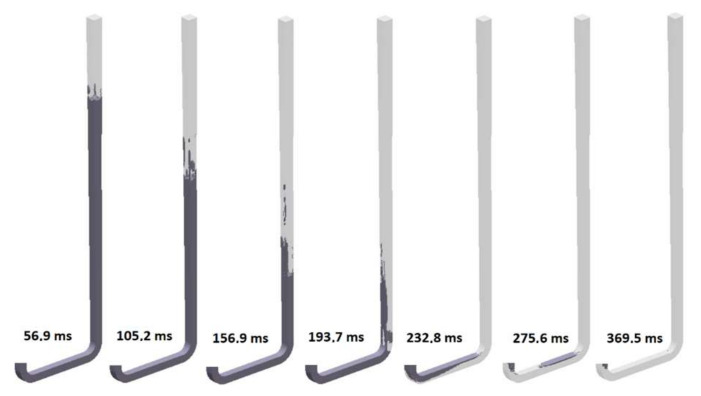
Air evacuation in square sprue with hyperbolic taper (S_s_7), MAGMASOFT.

**Figure 31 materials-15-04937-f031:**
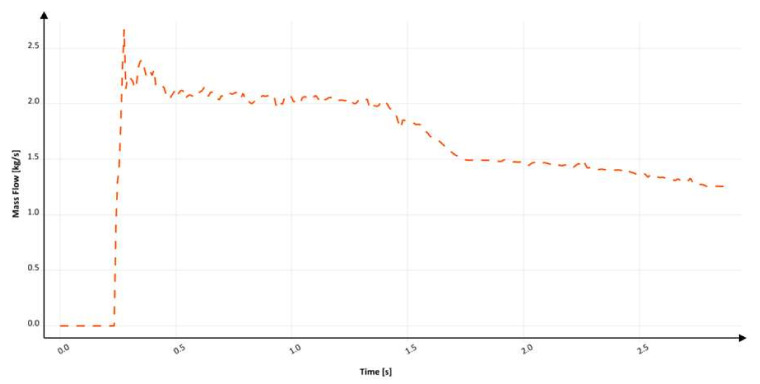
Mass flow through ingate with square sprue with hyperbolic taper (S_s_7), MAGMASOFT.

**Figure 32 materials-15-04937-f032:**
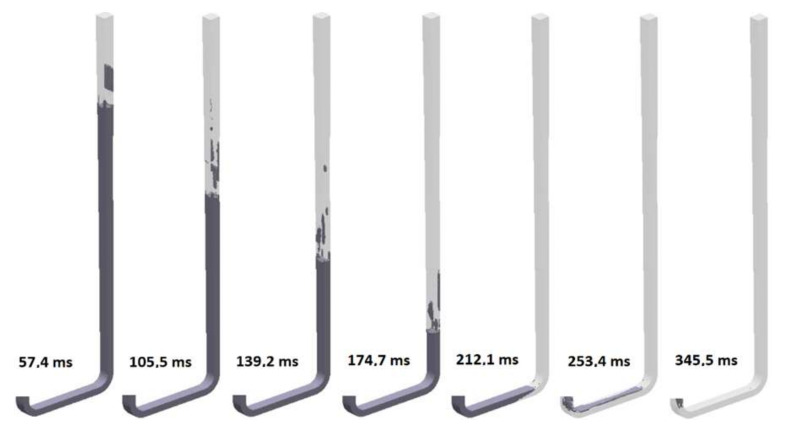
Air evacuation in thick rectangular sprue with hyperbolic taper (S_s_8), MAGMASOFT.

**Figure 33 materials-15-04937-f033:**
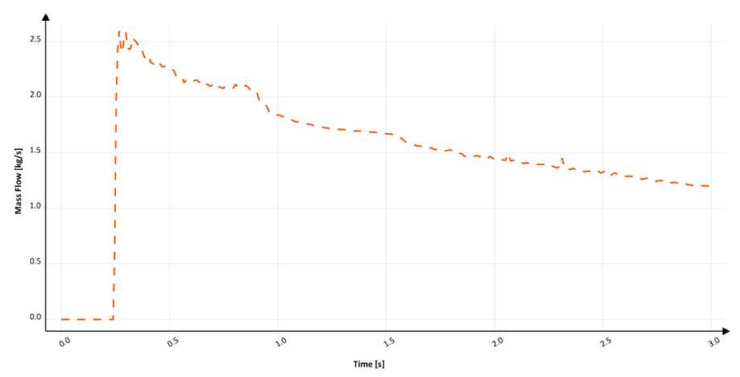
Mass flow through an ingate with thick rectangular sprue with hyperbolic taper (S_s_8), MAGMASOFT.

**Figure 34 materials-15-04937-f034:**
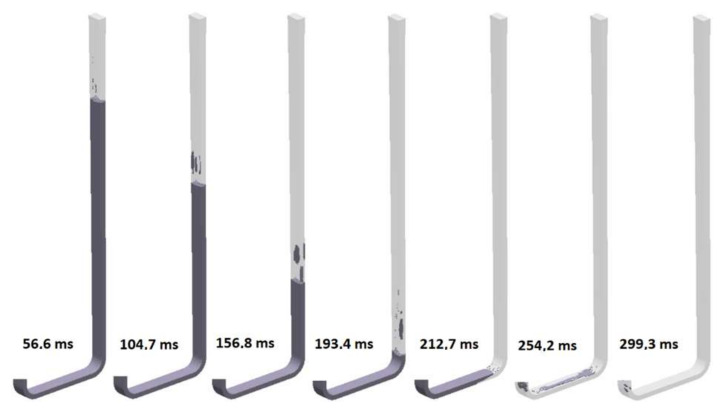
Air evacuation in thin rectangular sprue with hyperbolic taper (S_s_9), MAGMASOFT.

**Figure 35 materials-15-04937-f035:**
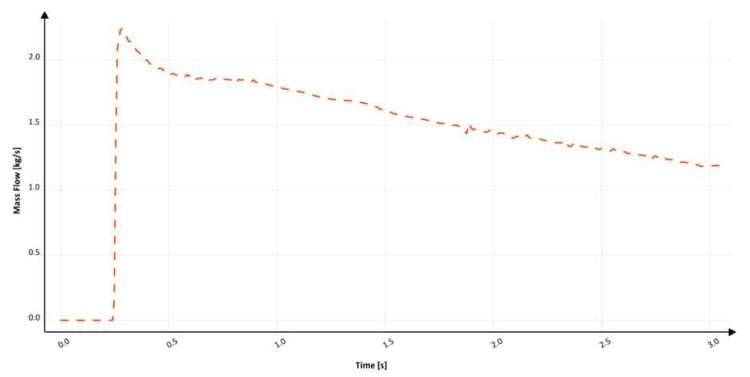
Mass flow through an ingate with thin rectangular sprue with hyperbolic taper (S_s_9), MAGMASOFT.

**Figure 36 materials-15-04937-f036:**
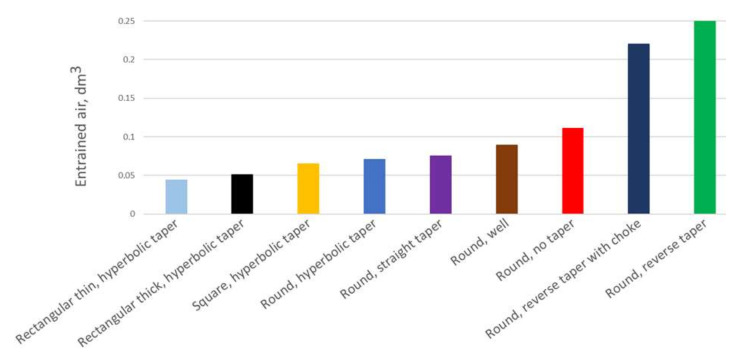
Air entrapment criterion for examined 500 mm sprues, MAGMASOFT.

**Figure 37 materials-15-04937-f037:**
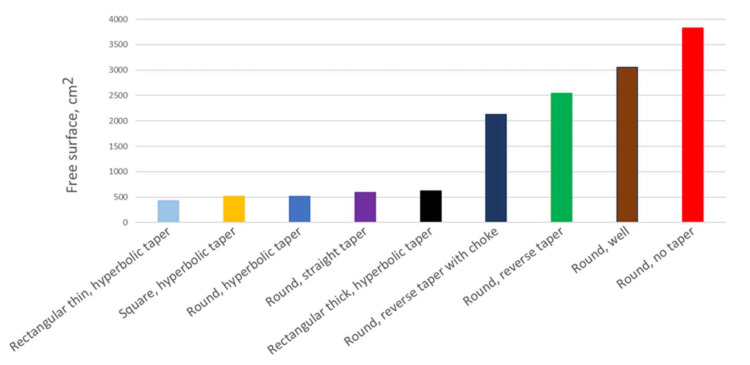
Smooth filling criterion for examined 500 mm sprues, MAGMASOFT.

**Figure 38 materials-15-04937-f038:**
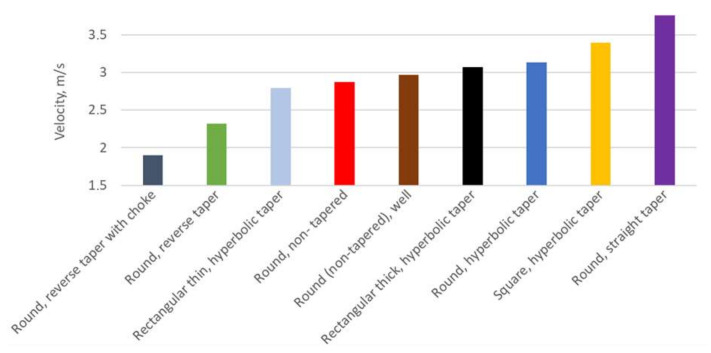
Average velocity criterion for examined 500 mm sprues, MAGMASOFT.

**Table 1 materials-15-04937-t001:** Sprue designs analyzed in initial studies.

Design No.	Sprue Type	Initial Surface Area	Base Surface Area
S_i_1	Round with no taper	333 mm^2^ (Ø 20.6 mm)	336,4 mm^2^ (Ø 20.7 mm)
S_i_2	Round with straight taper	333 mm^2^ (Ø 20.6 mm)	180 mm^2^ (Ø 15.1 mm)
S_i_3	Round with hyperbolic taper	333 mm^2^ (Ø 20.6 mm)	180 mm^2^ (Ø 15.1 mm)
S_i_4	Transitional with proper taper	333 mm^2^ (Ø 20.6 mm)	180 mm^2^ (30 mm × 6 mm)
S_i_5	Transitional with improper taper	333 mm^2^ (Ø 20.6 mm)	180 mm^2^ (30 mm × 6 mm)

**Table 2 materials-15-04937-t002:** Simulation parameters of initial studies of sprues.

Alloy	GS-52 According to DIN 1681
Pouring temperature	1570 °C
Molding sand	Alphaset
Mesh cell size	1.5 × 1.5 × 1.5 mm^3^
Height of the sprue	500 mm
Initial surface of the stream	333 mm^2^
Initial hydrostatic pressure	200 mm
Filling time	Resultant

**Table 3 materials-15-04937-t003:** Analyzed sprue designs with a length of 500 mm.

Design No.	Sprue Type	Initial Surface Area	Base Surface Area
S_S_1	Round, non-tapered	259 mm^2^ (Ø 18.2 mm)	259 mm^2^ (Ø 18.2 mm)
S_S_2	Round, reverse taper with choke	104 mm^2^ (Ø 11.5 mm)	138 mm^2^ (Ø 13.3 mm)
S_S_3	Round, reverse taper	104 mm^2^ (Ø 11.5 mm)	138 mm^2^ (Ø 13.3 mm)
S_S_4	Round, non-tapered, well	259 mm^2^ (Ø 18.2 mm)	259 mm^2^ (Ø 18.2 mm)
S_S_5	Round, straight taper	259 mm^2^ (Ø 18.2 mm)	138 mm^2^ (Ø 13.3 mm)
S_S_6	Round, hyperbolic taper	259 mm^2^ (Ø 18.2 mm)	138 mm^2^ (Ø 13.3 mm)
S_S_7	Square, hyperbolic taper	259 mm^2^ (Ø 18.2 mm)	138 mm^2^ (11.8 mm × 11.8 mm)
S_S_8	Thick rectangular, hyperbolic taper	259 mm^2^ (Ø 18.2 mm)	138 mm^2^ (9.2 mm × 15 mm)
S_S_9	Thin rectangular, hyperbolic taper	259 mm^2^ (Ø 18.2 mm)	138 mm^2^ (6.9 mm × 20 mm)

**Table 4 materials-15-04937-t004:** Simulation parameters of short sprue studies.

Alloy	EN-GJS-500-7 According to PN-EN-1563:2018-10
Pouring temperature	1360 °C
Molding sand	Furan
Mesh cell size	1 × 1 × 1 mm^3^
Height of the sprue	500 mm
Initial surface area of the stream	Equal to the initial area of the sprue
Initial hydrostatic pressure	200 mm
Filling time	Resultant

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
