# Peer review of "The Importance of the Geometry of the Down Sprue in the Gravity Casting Process"

_materials, 2022, doi:10.3390/ma15144937_

Round 1

Reviewer 1 Report

The authors have studied the importance of runner systems and their effect on the casting soundness. In many casting operations in foundry floor, the runner systems are not considered to be key parameter in cast parts quality. Direct pouring into the mould cavity is chosen as the simplest way of casting. In this work, the contribution to the field is so significant that the authors have put out a clear solid results that show the importance of runner design in producing defect free cast parts. They have also used simulation tools which aids the researchers and eliminating the need for trial-error. The experimental findings were supported by the simulation studies which contributes value information to the researchers who are working in this field. I found the article a well read, well organised where conclusions are supported and justified with relevant analytical tools.

Author Response

Dear Reviewer,

Thank you for your revision and the remarks. Please find in the attached file our responses for your revision. We did our best to incorporate the changes you requested.

Reviewer 2 Report

 1. Arrange the manuscript as Introduction, Materials & Method, Results & Discussion and Conclusion.

2. Avoid Figures in Introduction Part.

3. References are given in the list, But not cited in the manuscript.

4. The authors have used MAGMASOFT for analyzing. Give the boundary condition, element details etc.

5. The obtained results should be discussed with recently published articles.

Author Response

(The authors gave the same response as above.)

Reviewer 3 Report

This is an important modeling study, relying on the use of MAGMASOFT version 5.0 for predicting the amounts of air entrainment, filling times, and flow resistances, for various sprue systems connecting to a square mold cavity. The effects of rectangular, vs square, vs round sprues, for non-tapered, straight tapered, and hyperbolically tapered, sprue systems, are considered and rated by the authors based on the software available to them.

As such, it is important to understand the limitations of this software, in coming to firm conclusions. I would suggest that the authors consider the accuracy of MAG-MASOFT vs say FLUENT-ANSYS software, in simulating two-phase (gas-liquid metal) flows, using the VOF (Volume of Fluid), approach used there. I believe they should point out the limitations in terms of a macroscopic-based software, vs a microscopic-based software prediction, re bubble sizes, and two-phase flows. It would also help in presenting the results of all this modeling work presented in Figures 18 to Fig 36, with the adjective "Predicted; air evacuation, or predicted mass flow, or predicted air entrapment criterion, or predicted flow tracer....................: MAGMASOFT 5.0". 

A few spelling errors; In the introduction, change scrape to scrap the casting, Table 2 change Filing Time, to Filling Time, L 184 Change Cold Shots to Cold Shuts, L 229  change straight - taped, to straight-tapered"

Author Response

(The authors gave the same response as above.)

Reviewer 4 Report

This paper presents the results of experiments on the optimization of the downsprue geometry in the process of pouring of the sand molds. The best sprue geometry is selected from the point of view of minimization of the gas entrapment problem. However, there are still some problems in the paper, which should be revised and published.

1. The semi-industrial tests results were not given in the manuscript, it did not consistent with the introduction of the abstract, which should be suppled in the manuscript.

2. How to validate the simulation model based on the experimental results?

3. The quantity and quality of mesh have a great impact on the results of numerical simulation. No mesh-independence study was presented.

4. In Table 2 and Table 4, the unit of mesh cell size needs to be modified to 1.5 × 1.5 × 1.5m3. Meanwhile, the unit of initial hydrostatic pressure needs to be corrected.

5. In Figure 14, 3D model of the properly tapered sprue used in initial studies (right) needs to be supplemented with dimension information. In Figure 24, flow tracer in the sprue well needs to supplement time information.

6. The quality of Figures 15 should be improved, and more details should be described and added the scale.

7. Conclusion 4 holds that tapering allows to achieve a significant increase in yield, in performed examination, an increase of up to 60 % was obtained. “tapering” is a qualitative statement and “60 %” is a quantitative statement. This expression about a qualitative statements leading to a quantitative statement is not rigorous. This expression is recommended for revision.

Author Response

(The authors gave the same response as above.)

Round 2

Reviewer 4 Report

1. How the amount of trapped air was calculated in different cases?  We have not found any source of calculation for this data.

2. The Table 5 and the Figure 36 have the same meaning.

3. Some expression is incorrect just as in Figure 83 in line 646. The Figure 83 does not exist.

4. Maybe it is better to sum up all the curves of the mass flows through the ingates with different types sprues to compare the results.

Author Response

Dear Reviewer,

We have corrected the manuscript due to your remarks. Thank you for them all.

Sincerely,

Authors
